# Sanfilippo Syndrome: Molecular Basis, Disease Models and Therapeutic Approaches

**DOI:** 10.3390/ijms21217819

**Published:** 2020-10-22

**Authors:** Noelia Benetó, Lluïsa Vilageliu, Daniel Grinberg, Isaac Canals

**Affiliations:** 1Department of Genetics, Microbiology and Statistics, Faculty of Biology, University of Barcelona, CIBERER, IBUB, IRSJD, E-08028 Barcelona, Spain; noeliabg92@gmail.com (N.B.); lvilageliu@ub.edu (L.V.); dgrinberg@ub.edu (D.G.); 2Stem Cells, Aging and Neurodegeneration Group, Department of Clinical Sciences, Neurology, Lund Stem Cell Center, Lund University, SE-22184 Lund, Sweden

**Keywords:** Sanfilippo syndrome, mucopolysaccharidosis III, lysosomal storage disorders, heparan sulfate, animal models, induced pluripotent stem cells, cellular models, therapeutic approaches

## Abstract

Sanfilippo syndrome or mucopolysaccharidosis III is a lysosomal storage disorder caused by mutations in genes responsible for the degradation of heparan sulfate, a glycosaminoglycan located in the extracellular membrane. Undegraded heparan sulfate molecules accumulate within lysosomes leading to cellular dysfunction and pathology in several organs, with severe central nervous system degeneration as the main phenotypical feature. The exact molecular and cellular mechanisms by which impaired degradation and storage lead to cellular dysfunction and neuronal degeneration are still not fully understood. Here, we compile the knowledge on this issue and review all available animal and cellular models that can be used to contribute to increase our understanding of Sanfilippo syndrome disease mechanisms. Moreover, we provide an update in advances regarding the different and most successful therapeutic approaches that are currently under study to treat Sanfilippo syndrome patients and discuss the potential of new tools such as induced pluripotent stem cells to be used for disease modeling and therapy development.

## 1. Introduction

Lysosomal storage disorders (LSDs) comprise a heterogeneous group of rare inherited metabolic diseases that are characterized by the accumulation of macromolecules inside lysosomes. LSDs are caused by deficiencies in lysosomal enzymes, leading to lysosomal dysfunction, altered recycling of macromolecules, and impaired flux of the endolysosomal system. Mucopolysaccharidoses (MPS) are a group of LSDs accounting for approximately 30% of all LSD cases and arise from mutations in genes involved in glycosaminoglycans (GAGs) degradation, which accumulate inside the lysosomes [1]. Among MPS, Sanfilippo syndrome (also known as mucopolysaccharidosis III or MPS III) is the most frequent type and it was first described more than 50 years ago [2]. Sanfilippo syndrome is caused by mutations in the enzymes responsible for the degradation of heparan sulfate (HS), a specific GAG, and patients are characterized by severe neurological pathology leading to childhood dementia [3]. The role HS has in the development of the central nervous system (CNS) [4] can explain the severe neurological pathology found in patients. Moreover, in the last years, many studies have revealed the importance of the lysosomal system for maintaining neuronal and brain homeostasis, and alterations in the lysosomal system in many age-related neurodegenerative diseases [5]. Importantly, a link between Sanfilippo syndrome and Parkinson’s disease was suggested when mutations causing Sanfilippo syndrome were linked to a higher risk of developing Parkinson’s disease and aggregates of alfa-synuclein were found in patient brains [6]. For this reason, a better understanding of Sanfilippo syndrome’s underlying mechanisms can contribute to improve our knowledge on the role of impaired lysosomal function in age-related neurodegenerative disorders. Here, we revise the molecular basis of Sanfilippo syndrome from causative mutations to HS accumulation, and we summarize the latest advancements in therapeutic approaches as well as available animal and cellular disease models that can be used for investigating underlying mechanisms and assay potential therapies.

## 2. Sanfilippo Syndrome

There are four different subtypes of Sanfilippo syndrome based on the mutated gene and the consequent enzyme deficiency: type A (OMIM#252900), type B (OMIM#252920), type C (OMIM#252930), and type D (OMIM#252940), all of them presenting an autosomal recessive inheritance pattern [3]. Insufficient or complete loss of activity of any of the Sanfilippo syndrome causative enzymes leads to accumulation of partially degraded HS chains within lysosomes of cells in several organs and tissues [1,3,7]. In a recent study, a fifth subtype was identified in a mouse model [8] caused by mutations in the *ARSG* gene; however, to date, no human cases have been described. Moreover, human patients with a homozygous mutation in *ARSG* present Usher syndrome, leading to deaf-blindness and a small increase in urinary GAGs, although not as dramatic as in Sanfilippo syndrome patients [9].

Clinical symptomatology of Sanfilippo patients is similar regardless of the subtype, mainly characterized by an early-onset, severe, and progressive degeneration of the CNS with mild somatic symptoms [1,3,7]. Neurodegeneration starts during the first decade of life, with cortical atrophy, progressive dementia, motor deterioration, hyperactivity, learning difficulties, aggressive behavior, sleeping problems, and pronounced mental retardation [3]. Mild somatic manifestations include hirsutism, hepatosplenomegaly, joint stiffness, dysphagia, hypertrichosis, hypoacusia, speech loss, and skeletal alterations [1]. Death usually occurs at the second or third decade of life, although in unusual attenuated cases, life expectancy extends until the fifth or sixth decade [10,11,12,13,14].

The incidence of Sanfilippo syndrome varies depending on the subtype and geographical region, but on average is around one in 70,000 live births [15]. However, this incidence may underestimate the actual prevalence of different MPS III types because of the difficulties in the correct diagnosis of mild forms. Prevalence of the different subtypes vary between populations; subtype A being more frequent in the Northern Europe and subtype B more frequent in Southern Europe [16]. On the other hand, subtype C is in general less common while subtype D is very rare in all populations.

### 2.1. Subtype A

MPS IIIA or Sanfilippo syndrome type A is caused by mutations in the *SGSH* gene, coding for sulfamidase (also known as heparan sulfate sulfatase or N-sulfoglucosamine sulfohydrolase, EC 3.10.1.1), which releases sulfate groups linked to the amino group of glucosamine. The gene is localized at 17q25.3 [17] with an approximated length of 11 Kb and contains eight exons. It codes for a protein of 502 amino acids with five possible glycosylation sites and a total of 155 identified mutations (Table 1). Sanfilippo syndrome type A is considered the most aggressive form, with patients surviving until 15–18 years old on average [16].

### 2.2. Subtype B

MPS IIIB or Sanfilippo syndrome type B is caused by mutations in the *NAGLU* gene, which encodes N-acetyl-α-glucosaminidase (EC 3.2.1.50), a lysosomal enzyme of 720 amino acids with six possible glycosylation sites. The function of the enzyme is the hydrolysis of the linkage between N-acetylglucosamine (GlcNAc) and the uronic acid, the two saccharides that conform HS. The gene maps to 17q21.2 [18]; spans 8.3 Kb; contains six exons; and, to date, 229 mutations have been identified as shown in Table 1. Sanfilippo syndrome type B patients die on average between 17–19 years old, this subtype being slightly less aggressive than subtype A [16].

### 2.3. Subtype C

Mutations in the *HGSNAT* gene are responsible for MPS IIIC or Sanfilippo syndrome type C. This gene codes for the lysosomal membrane protein known as acetyl-CoA α-glucosaminide N-acetyltransferase (EC 2. 3.1.78). It is located at chromosome 8p11.1, was identified by two independent groups in 2006 [19,20], spans about 62.5 Kb, containing 18 exons, and gives rise to a protein of 635 amino acids. For some time, there was controversy about the real initiation codon [21], but a recent publication suggested that only one ATG codon worked as the initiation codon [22]. Until now, 77 mutations have been identified (Table 1). Subtype C is the less aggressive form of Sanfilippo syndrome, with a mean survival of 19–34 years depending on the study [16].

### 2.4. Subtype D

Mutations in the *GNS* gene, which encodes the lysosomal enzyme N-acetylglucosamine-6-sulfatase (EC 3.1.6.14), are responsible for MPS IIID or Sanfilippo syndrome type D. The gene is located at 12q14.3, is 46 Kb-long, and contains 14 exons. The enzyme has 552 amino acids and 13 potential glycosylation sites [23]. It catalyzes the sulfate removal in the N-acetylglucosamine residues. Until now, 25 mutations have been found (Table 1). Due to the rarity of this subtype, there is no data on average survival of patients.

## 3. Heparan Sulfate

HS is one of the most common and important GAGs located in the extracellular matrix as a part of proteoglycans in most animal tissues. HS is composed of many disaccharide units, comprised of glucuronic acid (GlcA) and GlcNAc that can be modified [24]. HS chains can differ depending on the composition and percentage of each modified disaccharides, giving rise to different types of HS and influences the biological activity of the HS chains [25]. The HS chains formed by these disaccharides repetitions are attached to a core protein being part of HS proteoglycans (HSPGs). A great variety of HSPGs can be found in the cell surface and extracellular matrix, such as syndecans, glypicans, or perlecans, based on the core protein and the type and number of HS chains linked to it [24]. HSPGs participate in many different cellular functions and systems such as cell migration, vesicle secretion system, endocytic system, cell adhesion and motility, membrane basement structure, and recognition of different factors and molecules as receptors or coreceptors [25].

### 3.1. Heparan Sulfate Biosynthesis

For HSPGs synthesis (Figure 1), it is first essential to form the different core proteins in the endoplasmic reticulum (ER). The number of these proteins, which could compete for HS synthesis, is the limiting factor in the synthetic pathway of different HSPGs. When the core protein is formed, the synthesis of the linkage region takes place from a specific serine residue next to a glycine and flanked by acidic and hydrophobic residues. This linkage region is formed by a xylose that binds the serine and is followed by two galactoses and one glucuronic acid. Although HS synthesis per se, is widely accepted to take place in the Golgi apparatus, the first enzyme involved in the formation of the linkage region acts in the ER. This linkage region is common to HS, chondroitin sulfate, dermatan sulfate, and heparin [24].

After the linkage region synthesis, HS chains are elongated (Figure 1). The elongation process starts with the addition of one N-acetyl glucosamine (GlcNAc) to the linkage region, step under control of the *EXTL2* and *EXTL3* genes, whose products (EXTL2 and EXTL3) have GlcNAc-transferase I activity [24]. After this initiation step with the participation of the *EXTL* genes, the elongation of the HS chain takes place by the action of the EXT1–EXT2 complex, which alternatively adds glucuronic acid (GlcA) and GlcNAc residues to the chain, forming polymers of different length [26]. Mutations in *EXT1* and *EXT2* result in reduced HS in the cartilage and cause hereditary multiple exostoses, an autosomal dominant disorder affecting skeleton with a risk of malignant transformation [27].

During the synthesis of the HSPGs, HS chains can be modified (Figure 1) through the activity of six different enzymes (sulfotransferases, deacetylase, and epimerase) resulting in deacetylation of some GlcNAc, sulfation of deacetylated GlcNAc, epimerization of some GlcA to form iduronic acid (IdoA), and sulfation of some IdoA and GlcA residues [25]. All of these modifications are important for HS interactions and recognition of different factors and molecules.

### 3.2. Heparan Sulfate Degradation

HS is degraded within the lysosomes (Figure 1), to which it arrives through the endosomal pathway [26]. First, in the extracellular matrix, some endosulfatases and a secreted heparanase could partially degrade HS chains, giving rise to smaller fragments. However, final HS degradation takes place inside the lysosomes after internalization of HSPGs through the stepwise action of nine different enzymes [1]. The first enzyme of the pathway, heparanase, is an endoglucuronidase that cleaves HS chains into smaller fragments to facilitate the polymer degradation.

After the fragmentation, the following enzymes proceed with the complete degradation of the small fragments by acting sequentially (Figure 1): iduronate 2-sulfatase (IDS), α-L-iduronidase (IDUA), heparan N-sulfatase or sulfamidase (SGSH, mutated in Sanfilippo syndrome type A), acetyl-CoA α-glucosaminide N-acetyltransferase (HGSNAT, mutated in Sanfilippo syndrome type C), α-N-acetylglucosaminidase (NAGLU, mutated in Sanfilippo syndrome type B), glucuronate 2-sulfatase (GDS), β-glucuronidase (GUSB), and N-acetylglucosamine 6-sulfatase (GNS, mutated in Sanfilippo syndrome type D) [26]. It has been suggested that all these enzymes function as a complex in the lysosomes [28].

### 3.3. Heparan Sulfate Accumulation and Disease Mechanisms

HSPGs are essential components of the cell surface and extracellular matrix, providing structural support to glial and neuronal cells. Importantly, they regulate several signaling pathways; control the proliferative capacity of neural progenitors; are essential for brain patterning and neurogenesis; and crucially, participate in the processes of neuronal migration, axon guidance, and synaptogenesis [29,30,31]. The fact that the CNS has a limited capacity of regeneration, a high sensitivity to damage, and a need of long cellular survival could explain the severe neural pathology in Sanfilippo patients both at the CNS and the peripheral nervous system. Moreover, the important roles of HSPGs in CNS development suggest that it can be of interest to investigate early neurodevelopmental alterations in disease models that can shed light into new disease mechanisms leading towards the severe neurological pathology found in patients.

HS accumulation causes an alteration in the lysosomal environment since the excess of undegraded molecules can bind to various hydrolases reducing their activity [32] and causing secondary accumulation of gangliosides and other GAGs within and outside of the lysosome that may contribute to the CNS pathology [4]. Moreover, HS storages are found not only within the lysosome but also in other subcellular locations, affecting the CNS functionality [33]. In addition, the storage of undegraded molecules affects intracellular trafficking and the flux in the endolysosomal and autophagic pathways [4,34]. It is possible that HS fragments released to the extracellular matrix interfere with many HS functions, favoring disease development. HSPGs act as ligands for several factors such as FGF and BMP4, whose signaling is affected due to disease-related imbalanced turnover of HSPGs [4].

Within the CNS, the increase in HS constitutively activates focal adhesions in glial and neural cells and impairs polarization and migration of neurons [35]. Importantly, and considering the role of HSPGs in modulating cellular immune signaling, the injury in neurons, and the constant release of inflammatory mediators, a clear inflammation and microgliosis have been found due to HS-related CNS alterations in several animal models [36,37,38,39]. In a mouse model of Sanfilippo B disease, a clear accumulation has been found in storage vesicles within microglial cells [40]. This cell type is essential for the brain’s defense, and its alteration may lead to a release of toxic products that, together with the neuroinflammation, may contribute to neurodegeneration in Sanfilippo syndrome patients.

Another disease mechanism that has been described is a consequence of a secondary storage of gangliosides in Sanfilippo syndrome. This secondary accumulation is due to a failure of several lysosomal enzymes caused by the change in the lysosomal environment as a consequence of the primary storage. Storage of gangliosides has been shown to lead to reduced uptake of calcium by the ER, together with a consequent increase of cytosolic calcium levels that trigger neuronal apoptosis, thus favoring neurodegeneration. Decreased ER calcium can activate the unfolded protein response, which also triggers apoptosis [41] and contributes to the severe neurodegeneration.

On the other hand, in a very interesting work with a mouse model of multiple sulfatase deficiency, it was shown that animals carrying the mutation only in the astrocyte lineage were also developing neurodegeneration [42]. This work was the first evidence that astrocyte dysfunction can be sufficient to trigger neuronal degeneration in lysosomal storage disorders, although with slower progression than animals with all brain cells carrying the mutation. Importantly, the role of astrocytes in neurodegeneration has now been proved for other neurological disorders [43].

Moreover, in an MPS IIIA mouse model, a high number of autophagosomes were found as a consequence of the impairment in the autophagy–lysosomal function, which probably leads to cell death [44]. Altogether, there are several functions affected due to HS accumulation and impairments are not only found in neurons but in other brain cells, although studies have been mainly done on animal models.

Considering the higher complexity of the human brain and human neural cells compared to murine counterparts, it is crucial to generate relevant cellular models to investigate human disease mechanisms.

## 4. Disease Models

Animal and cellular models are essential in order to allow for studies on the underlying mechanisms of disease and to assess potential therapeutic approaches before they can be moved into clinical trials. In neurodegenerative disorders, for many years, animal models represented the best tool for these purposes considering the obvious difficulties in obtaining human brain cells. However, in the last years, the development of the induced pluripotent stem cells (iPSCs) technology has facilitated access to a constant source of patient-derived neural cells. This milestone has been especially important for the study of disorders affecting the CNS. Here, we will briefly summarize the existing animal models and their phenotypic resemblances to human patients and, posteriorly, we will examine the iPSC-derived models of Sanfilippo syndrome.

### 4.1. Animal Models

Three different natural animal models have been described for Sanfilippo A syndrome, a Dachshund dog [45], a Huntaway dog [46], and a mouse [47]. All these animal models mimic the human phenotype, with a progressive neurodegeneration, loss of motor abilities and mild somatic symptoms such as hepatosplenomegaly, and the urinary excretion of GAGs. Impairment in the autophagy function has also been detected in the mouse model [44]. Due to the fact that natural mouse models were identified in a mixed background, another study developed a new congenic strain to ensure long-term stability and genetic homogeneity of the model. Its characterization resulted in minor differences between different MPS IIIA mouse strains [48].

Three natural animal models were described for Sanfilippo B syndrome, a Schipperke dog [49], an avian model [50], and a herd of cattle carrying a mutation previously described in human patients [51]. These natural animal models presented similarities with the human disease, such as motor deterioration, low enzyme activity, and storage of GAGs in different tissues. Furthermore, one mouse model has been generated [40] by the disruption of exon 6 of the mouse orthologous *Naglu* gene. These mice presented some symptomatology that mimics the human disorder, such as HS accumulation in different tissues, vacuolization in different cell types, secondary storage of gangliosides, hearing loss, and a shorter lifespan.

No natural models have been found for MPS IIIC, but two mouse models of Sanfilippo C have been recently generated [36,52] by *Hgsnat* disruption. In both cases, animals showed typical phenotypic alterations related to Sanfilippo syndrome such as hyperactivity, motor dysfunction, and cognitive decline together with common cellular defects such as lysosomal accumulation of HS, enlarged vesicles, mitochondrial impairments, microglial activation leading to neuroinflammation, and neuronal loss.

One natural animal model was described for Sanfilippo type D, a Nubian goat [53]. Recently, a mouse model of MPS IIID was also generated using mouse embryonic stem cells carrying a disruptive insertion in the murine *Gns* gene [54]. Both models presented neurological and histological manifestations similar to those of human patients.

### 4.2. Cellular Models

For a long time, reliable cellular models to study the neurodegenerative aspects of Sanfilippo syndrome were lacking. Primary cells are difficult to obtain for obvious ethical reasons and when available, they come from post-mortem material, which represents the late stages of the disease and do not allow for investigating early mechanisms leading to neurodegeneration. Fibroblasts and HeLa cells have been largely used but the essential morphological and functional differences compared to brain cells make them a weak model to investigate the molecular basis of this neurodegenerative disorder. Some studies used animal-derived stem cells or brain cells which, as mentioned for animal models, present a lower degree of complexity when compared to human counterparts—especially astrocytes, the main glial cell in the brain. With the development of the iPSC technology by Yamanaka and collaborators in 2006 [55], a door for generating patient-specific brain cells was opened. After reprogramming patient fibroblasts into iPSCs, several available differentiation protocols allow to generate neuronal and glial cells for disease modelling.

It was in 2011 when for the first time, iPSCs were used to generate a Sanfilippo B neuronal model [56]. Human iPSC lines were obtained through fibroblasts reprogramming using a control line and two *NAGLU*-mutated patient lines with different transcription factor combinations based on the Yamanaka factors. Interestingly, for patient cells, NAGLU enzyme supplementation was required to clear fibroblast HS storage in order to achieve a successful reprogramming process. HS storage-related lesions were detected already at the iPSC stage although neuronal differentiation was not affected. The expression profiles in neural stem cells (NSCs) pointed out differences in Golgi-related genes and also showed alterations in extracellular matrix genes. iPSC-derived neurons presented an increase in the amount of Golgi matrix protein GM130, which is essential for the proper function of the Golgi, thus indicating numerous abnormal organized Golgi complexes. Importantly and as expected, neurons presented a high number of LAMP1 and GM3 positive storage vesicles, thus confirming the known secondary storage of gangliosides in Sanfilippo syndrome patients. This was the first time that Golgi alterations were identified in a human model of Sanfilippo syndrome, however, it remains unclear whether the impairments in the Golgi function are a cause or consequence of the lysosomal dysfunction. Recently, a new work from the same lab has shown alterations in cell polarization and migration using this iPSC-based cellular model, which may contribute to the neurological phenotype of Sanfilippo patients [35].

Later on, we generated iPSCs from fibroblasts of two Sanfilippo C patients and a healthy control that were differentiated into neural cultures [57]. In that study, affected cells showed an increase in lysosome number and size together with a clear reduction in enzymatic activity and a consequent accumulation of GAGs. Moreover, affected lines displayed impaired neuronal network activity and connectivity, defects that could be restored by ectopic expression of *HGSNAT*. However, specific astrocyte alterations were not investigated, and the role of these essential cell types in disease development remains to be elucidated. 

Although no in vitro human neural model has been generated for Sanfilippo A, recently, it has been reported that the generation of two lines of iPSCs from fibroblasts of the same patient [58] could be used for the generation of these models. In the case of Sanfilippo B, new iPSC lines from patient fibroblasts have also been generated [59,60].

One of the major hurdles of using iPSCs is the difficulty to discriminate effects due to distinct genetic backgrounds of different cell lines. Recently, the development of the CRISPR/Cas9 system [61], an easily accessible and programmable RNA-based genome editing tool, has facilitated the generation of isogenic control lines from iPSCs, either to introduce specific mutations in a healthy line or to correct mutations in patient-derived lines. Isogenic control lines generated with CRISPR/Cas9 are essential to avoid detecting non-disease-related phenotypes arising from different genetic backgrounds of patient- and control-iPSC lines. Importantly, after genome editing, proliferation and differentiation capacity of iPSCs is not affected [62]. Therefore, very recently and to complement our previous Sanfilippo C syndrome iPSC lines, we have generated two HGSNAT-mutated iPSC lines from the healthy control line [63] through the use of the CRISPR/Cas9. In a posterior work with these iPSC lines, we generated induced neurons and induced astrocytes that recapitulated two main hallmarks of the disease—the loss of enzymatic activity and the accumulation of HS [64]. In addition, using the same CRISPR/Cas9 approach, we generated two Sanfilippo B syndrome iPSC lines from the same healthy control line [65]. Altogether, these iPSC lines will allow for studies on molecular and cellular disease mechanisms in specific brain cell types, find common alterations as well as particularities in different Sanfilippo syndrome subtypes, and assay different therapeutic strategies to treat these devastating disorders.

## 5. Therapeutic Approaches

Currently, there is no treatment to effectively slow down or reverse Sanfilippo syndrome patients’ neurodegeneration, and their management consists only of palliative measures to alleviate the symptomatology. Interestingly, different kinds of approaches have been tested during the last years in cellular and animal models of the disease, focused mainly on the treatment of the CNS involvement. The main approaches we will review here consist of enzyme replacement therapy (ERT), substrate reduction therapy (SRT), pharmacological chaperones, stem cell transplantation, and gene therapy (Figure 2). However, other approaches such as the use of coenzyme Q_10_ [66]; overexpression of TFEB [67], the master regulator in the lysosome biogenesis [68,69]; or the use of modified RNAs to recover aberrant splicing processes [70] have also been assayed showing different potentials to ameliorate pathological features of cellular and animal models.

### 5.1. Enzyme Replacement Therapy

The success of any therapy relying on administration or production of the correct form of the lysosomal enzyme relies on the fact that these proteins are tagged with mannose 6-phosphate (M6P) for correct trafficking towards the lysosome. Considering that cells have M6P receptors in the membrane, lysosomal enzymes can be endocytosed and arrive to the lysosome to perform their function [71]. For non-neurological LSDs, exogenous administration of the correct form of the enzyme mutated in patients, known as ERT (Figure 2A), has been proven to be the most successful strategy [72]. However, for diseases affecting the CNS, the existence of the blood–brain barrier (BBB), which limits the availability of the enzyme in the brain, has to be taken into account. In addition, antibodies targeting the enzyme can be observed in treated LSD-patients, clearly reducing the efficiency of the ERT [73]. Thus, intravenous administration is not as useful as for other LSDs without CNS pathology, for which ERT is currently approved and in use. On the other hand, direct brain administration for the treatment of neurological disorders seems more beneficial [74], although it is an aggressive treatment that needs continued injections. Nevertheless, clinical trials based on ERT for Sanfilippo syndrome type A and B have been carried out without clear results [75,76,77,78]. In any case, research to further investigate the potential of this approach is required [79].

### 5.2. Substrate Reduction Therapy

Taking into account the limitations of ERT, SRT has been presented as a valid alternative approach. The objective of this therapy is to find molecular targets to decrease the production of the accumulated substrate and restore the balance between synthesis and degradation (Figure 2B). It is important to remark that the mutant enzyme has to maintain some residual activity in order to achieve this restoration. SRT has been already approved to treat some LSDs, both with neurological and non-neurological symptomatology [80,81]. For Sanfilippo syndrome, different molecules with the ability to cross the BBB for the treatment of the CNS have been tested.

One of the most studied of these molecules is genistein, a natural isoflavone that inhibits the kinase activity of epidermal growth factor receptor, which is important for complete expression of genes encoding enzymes responsible for GAG production. Genistein was able to reduce GAG production in Sanfilippo syndrome type A and B fibroblasts [82], and to improve behavioral abnormalities, neuroinflammation, synaptic loss, and lysosomal storage in a Sanfilippo B mouse model [83]. After these positive results, two clinical trials with genistein treatment were carried out showing a reduction in urinary GAGs, but with unclear neurological benefits [84,85]. Another clinical trial using a higher dose of genistein was recently completed for Sanfilippo syndrome types A, B, and C. Even though these doses were safe for the patients, only a slight reduction of HS in the cerebrospinal fluid was observed, with no attenuation of the intellectual disability [79]. Further studies with higher doses of genistein and other flavonoids should be carried out to establish the ability of this group of molecules to ameliorate CNS pathology in Sanfilippo patients. 

A different and interesting option for SRT is the use of specific RNAi directed to key genes involved in the GAG synthesis such as *EXTL* genes or genes involved in the linkage region formation. RNAi is a mechanism to selectively silence the expression of a particular gene by the specific degradation of the mRNA. Synthetic siRNAs and shRNAs have been widely used to downregulate the expression of a large number of genes in several cell types in vitro and in vivo. In one study, siRNAs were used to downregulate *XYLT1*, *XYLT2*, *GALTI*, and *GALTII*, genes encoding enzymes responsible for the formation of the linkage region [86] (Figure 1). This strategy was assessed in MPS I and MPS IIIA fibroblasts, resulting in an important decrease at the mRNA and protein levels for all the genes and a consequent significant decrease in the GAG synthesis after three days of treatment. In another study performed in our lab, the use of shRNAs to downregulate *EXTL2* and *EXTL3* genes was found to reduce the GAG synthesis and storage in MPS IIIA fibroblasts. These results were observed after three days of treatment, but failed after seven days [87]. Later on, fibroblasts from Sanfilippo C patients treated with similar siRNAs targeting *EXTL2* and *EXTL3* genes showed a reduction in GAG synthesis after three days and a decrease in HS storage after two weeks [88]. However, these studies were performed on patients’ fibroblasts, therefore, it is important to study SRT in relevant human neural cells, which are the ones affected in patients. In a recent study, we demonstrate that the same siRNAs that were effective in Sanfilippo syndrome type C fibroblasts were not efficient in decreasing storage in iPSC-derived neurons generated from the same fibroblasts assayed in the previous study [64].

### 5.3. Pharmacological Chaperones for Enzyme-Enhancement-Therapy

In many cases, missense mutations lead to the production of misfolded proteins that are rapidly degraded due to misfolding but that conserve some residual activity [89,90] and chaperons are cellular proteins that help proteins to adopt correct foldings. For years, several small compounds that act as chaperones, preventing misfolding of mutant proteins, have been identified (Figure 2C). Among the most common pharmacological chaperones that have been used for enzyme-enhancement therapy are amino and iminosugars. These molecules are in fact enzyme inhibitors that interact specifically with the active site of proteins and, used at low concentrations, can effectively stabilize the mutant enzymes and restore the correct folding to facilitate their trafficking towards the lysosome, thus, partially restoring enzymatic activity. In the case of LSDs, it has been proposed that achieving an enzyme activity around 5–15% can be sufficient to avoid the appearance of pathological symptoms [91]. To date, several compounds with chaperone activity have been tested for different LSDs such as Fabry disease, G_M1_-gangliosidosis, Morquio B disease, Pompe disease, Gaucher disease, Krabbe disease, Niemann-Pick A/B and C diseases, as well as for other types of disorders such as retinitis pigmentosa, cystic fibrosis, Parkinson’s disease, Alzheimer disease, or cancer [92]. 

In the case of Sanfilippo syndrome, several compounds were tested in a study for their potential to act as pharmacological chaperones [93]. The results showed that glucosamine, a competitive inhibitor of the *HGSNAT* enzyme with low toxicity, significantly increased HGSNAT activity in most patient fibroblasts lines tested, indicating its therapeutic potential. For Sanfilippo syndrome type C, we carried out a preclinical cell-based study showing a 2.5-fold increase of HGSNAT enzyme activity using glucosamine in patients’ fibroblasts carrying one splicing mutation that produces a protein lacking four amino acids [70]. Further studies should be done in order to establish its efficacy and lack of toxicity in brain cells as well as its ability to cross the BBB.

### 5.4. Stem Cell Therapy

In the last few years, several stem cell applications have been described for the treatment of neurological diseases in order to deliver the correct form of the enzyme into the brain (Figure 2D). Allogeneic bone marrow transplantation is used in the treatment of different LSDs with neurological pathology, but in the case of MPS III, intravenous administration of lentiviral-transduced bone marrow stem cells were not efficient to treat a mouse model of MPS IIIA [94] due to an insufficient production of enzyme by the donor cells or an inefficient uptake by the host cells [95]. 

Hematopoietic stem cell transplantation has been largely tested in many patients suffering from different LSDs. In patients’ brain, these cells can replace microglia and become enzyme-secreting donor cells [96]. Nevertheless, this process seems to be slow and not complete, making this option an invalid therapy for neurological disorders with a rapid progress of symptoms such as Sanfilippo syndrome, and currently, this approach is no longer considered for the treatment of Sanfilippo syndrome [97]. However, recent works using genetically modified hematopoietic stem cells carrying the normal copy of the *SGSH* or *NAGLU* genes showed an improvement in the neurological pathology in MPS IIIA or MPS IIIB mouse models [98,99,100,101].

Administration of human umbilical cord blood cells to the MPS IIIB mouse model has been explored, resulting in an amelioration of the neurological and somatic symptoms [102]. However, it presents the inconvenience that the enzyme production declines with time. On the contrary, the transplantation of umbilical cord blood-derived stem cells in two type B patients before the disease onset did not prevent the neurological deterioration [103]. 

Direct cell transplantation in the brain can be useful to both serve as cell replacement therapy addressing neuronal loss, as well as a source of cells secreting the correct version of the deficient enzyme [104]. In the last years, the development of iPSC technology has allowed researchers to easily generate patient-specific neural stem cells (NSCs), which have the potential to give rise to neurons, astrocytes, and oligodendrocytes. After transplantation into murine brains, NSCs can migrate long distances within the brain, differentiate, and integrate in the host network without disrupting normal functionality. In conclusion, NSCs represent an extraordinary opportunity to distribute the wild-type (WT) lysosomal enzyme and to recover neurological pathology, as it has been shown in studies in which MPS VII [105] and MPS IIIB [106] mouse models were treated with this strategy. However, for Sanfilippo C it is important to consider that HGSNAT does not have a M6P tag and is a membrane protein, therefore secretion and uptake of this enzyme by deficient cells may not be successful.

The use of glial precursors cells (GPCs) derived from pluripotent stem cells is another potential therapy to treat LSDs. In the mouse model of MPS IIIA [107], GPCs genetically modified to overexpress the *SGSH* gene were tested. Results showed promising results for this therapeutic approach, with GPCs successfully engrafting and surviving in the host brain, not forming teratomas, and showing long-term *SGSH* overexpression. Interestingly, astrocyte-based therapies are emerging as an option to treat some neurodegenerative disorders in which astrocytes play important roles, such as amyotrophic lateral sclerosis [108].

### 5.5. Gene Therapy

Gene therapy consists of the delivery of the correct copy of the gene to affected cells in order to recover enzyme activity (Figure 2E). Gene therapy is the most promising therapeutic option for LSDs since, as already referred, it has been proposed that only 5–15% of enzyme activity is required to maintain a healthy condition in affected patients [72]. Several clinical trials are currently ongoing or scheduled for different MPS [79]. In the case of Sanfilippo syndrome, several viral vectors have been tested for their therapeutic potential, such as retroviruses, lentiviruses, adenoviruses, and adeno-associated viruses (AAV). In addition, an approach using a nonviral vector (pFAR4) via tail vein administration was shown to increase enzyme activity and reduce GAGs storage in several tissues and lysosomes in the brain of an MPS IIIA mouse model [109]. Importantly, authors showed that liver of treated animals was converted into an enzyme distributor that promoted the GAG decrease in other tissues.

In the last years, the use of AAV have become the gold-standard tool for gene therapy in neurological disorders. Among the qualities that make them good vectors, it is important to highlight that they are nonintegrative, nonpathogenic, and nonimmunogenic in humans, and have the capacity to infect nondividing cells providing long-term expression. However, a recent study shows that AAV can induce cell death in some neural cell types in the murine hippocampus, suggesting that these approaches should be carefully evaluated [110]. Nevertheless, in the last few years, several reports have been published concerning AAV-mediated therapy using different virus serotypes and delivery strategies. 

For MPS IIIA, intracerebral administration of AAV5 carrying the *SGSH* gene together with the *SUMF1* gene (coding for an essential and limiting factor for sulfatases) in the mouse model showed an increase in the SGSH activity in the brain, a decrease in the storage and inflammation, and an improvement in the motor and cognitive function [111]. After these results, a phase I/II clinical trial for MPS IIIA using AAV10 expressing the deficient SGSH enzyme and the SUMF1 enzyme was started. It recently finished, showing no toxicity or lack of tolerance and a possible slight improvement in patient behavior [112]. AAV5 has also been used in another clinical trial with MPS IIIB patients, and results indicate an improvement of neurocognitive progression in all patients [113].

AAVrh10 has also been used to deliver *SGSH* in MPS IIIA mice via intraparenchymal administration [114]. This treatment reduced HS and GM3 ganglioside accumulation and microglial activation, but only in the site of injection. To increase efficacy, multiple intraparenchymal regions should be injected to ensure widespread distribution. To study SGSH distribution in the brain of large animals, the same transducing vector was injected via parenchyma in dogs and cynomolgus monkeys, and SGSH enzyme activity increase was detected [115].

In a study comparing delivery efficiency of the *NAGLU* gene using different AAV serotypes in MPS IIIB mice [116], a better biodistribution and transduction was found using AAV8 via direct administration of the virus to the CNS, but AAV9 showed better results for systemic or intracerebroventricular delivery. Intramuscular administration of AAV8 carrying the *SGSH* gene in Sanfilippo A mouse models showed no amelioration, while intravenous administration was effective in transducing mainly the liver, with a consequent amelioration of the pathology in somatic tissues, although with a discrete improvement in CNS symptoms of male mice [117]. To improve secretion and targeting of the CNS, another study used a fusion protein of *SGSH* with a signal peptide to boost enzyme secretion and a BBB-binding domain. This vector was administered with an AAV8, and results showed an important increase in enzyme activity in the brain that resulted in brain pathology and behavior improvements [118].

Recently, the safety of intravenous administration of an AAV9 carrying the *NAGLU* gene was tested in unaffected primates [119]. AAV9 has been suggested to be the most efficient serotype for targeting brain cells and therefore, for the treatment of neurological disorders. Very interestingly, a consistent and long-term increase in brain enzymatic activity was detected together with low immunogenic reaction. Similar successful results using AAV9 have been achieved in mouse and canine models of MPS IIIA [120,121]. First, a clear increase in enzyme activity combined with a reduction in GAG storage and neuroinflammation was found in the mouse model treated intravenously, resulting in expanded lifespan [121]. Later, both animal models were treated with intracerebrospinal injections, showing low immunogenic reaction and resulting in a clear restoration of enzymatic activity and full body reduction of GAG storage and lysosome alterations, leading to prolonged lifespan [120]. This same research group also develop a strategy to treat MPS IIIB [122] or MPS IIID [54] mice with cerebrospinal fluid delivery of AAV9 vector carrying *NAGLU* or *GNS* genes, respectively. After treatment, enzyme activity in the CNS, normalization of GAG storage, corrected behavior, and extended lifespan were observed.

All these results in different Sanfilippo subtypes encouraged the application of this approach in human patients. In relation to cerebrospinal fluid administration, Esteve Laboratories recently started a phase I/II clinical trial using AAV9-*hSGSH* in MPS IIIA patients (EudraCT Number: 2015-000359-26). Besides, although some preclinical studies have been performed before, it was recently confirmed that some AAV were able to cross the BBB [123]. Due to that, Abeona Therapeutics has started a clinical trial using an intravenous delivery of AAV9 vector carrying the human *SGSH* gene under the control of a U1a promoter (ClinicalTrials.gov: NCT02716246, NCT04088734). Preliminary data showed a dose-dependent and sustained reduction in cerebrospinal HS after 30 days. In the case of Sanfilippo syndrome types A and B, two clinical trials based on intracerebral injection of AAV have been already completed [112,113], and another two for subtype A have started (ClinicalTrials.gov: NCT03612869, EudraCT Number: 2015-000359-26). However, as for ERT, gene therapy success for lysosomal enzymes relies in the ability of transduced cells to share the correct lysosomal enzyme through M6P receptors with non-transduced neighboring cells [68]. As mentioned above, HGSNAT is a lysosomal transmembrane protein that does not undergo the M6P pathway. For this reason, Sanfilippo C syndrome might not be the best candidate for gene therapy strategy, although some interesting results have been obtained with a novel AAV [124].

## 6. Conclusions

With this review, we provided an overview of the molecular basis of Sanfilippo syndrome and a summary of the main animal and cellular models available to date, which can be used to test the different therapeutic approaches. For many years, animal models have been the gold-standard to investigate disease mechanisms and to develop and test therapeutic options. However, the fundamental differences between animal and human brain structure and development, as well as the higher complexity of human organs and cells, are major factors that should be taken into account when performing a study. Findings relevant in animal models might not be important in the context of human physiology, pointing out the importance of generating human cellular models to complement the existing animal models. In the last years, several Sanfilippo-iPSC lines have been established and, in combination with faster and improved protocols to generate relevant cell types in 2D and 3D cultures [125,126,127], will contribute to expand our understanding of the molecular and cellular mechanisms of the disease. In addition, iPSC-derived brain cells will be very useful in drug screening studies to identify possible drug candidates with the potential to treat human brain cells. To date, several potential therapies have been tested, however gene therapy seems to be the approach generating a better outcome. It is crucial to follow the ongoing clinical trials for this therapy considering the very promising results of previous studies. Nevertheless, efforts should also be made to develop and assess other possible approaches for the treatment of patients suffering from this devastating disorder.

## Figures and Tables

**Figure 1 ijms-21-07819-f001:**
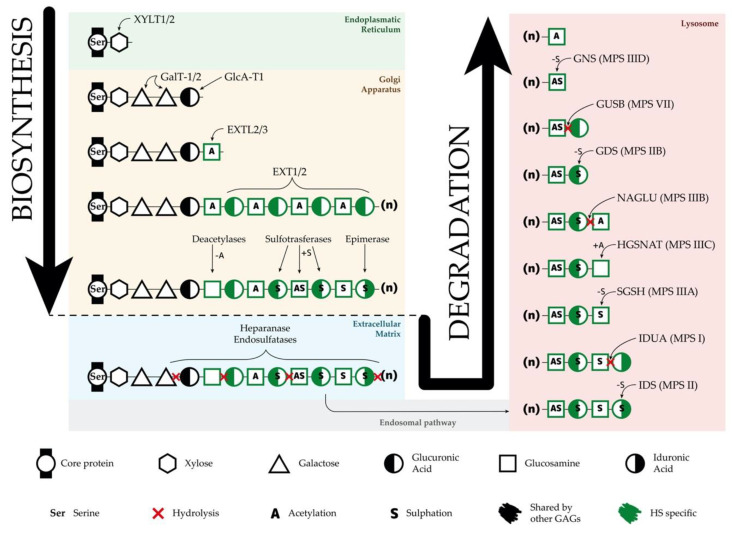
Synthesis and degradation of heparan sulfate (HS). Schematic representation of the biosynthesis and degradation processes of HS, including organelle location of each step, enzymes responsible for each function, residues in the HS chains, and modifications of these residues. GAGs—glycosaminoglycans.

**Figure 2 ijms-21-07819-f002:**
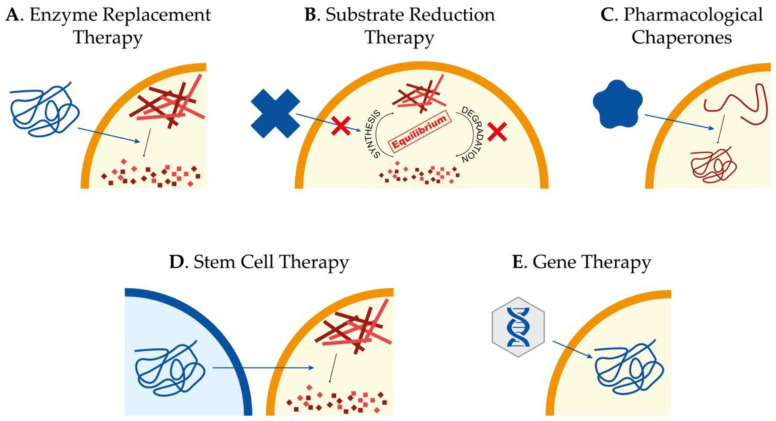
Potential therapeutic approaches to treat Sanfilippo syndrome. Schematic representation of the main therapeutic strategies currently being studied for the treatment of Sanfilippo syndrome patients: enzyme replacement therapy to provide the correct form of the mutated protein (**A**), substrate reduction therapy to reduce storage of undegraded molecules (**B**), use of pharmacological chaperones to correct protein missfolding (**C**), stem cell therapy for regeneration and production of the correct form of the protein (**D**) and gene therapy to provide cells with the correct form of the mutated gene (**E**).

**Table 1 ijms-21-07819-t001:** Distribution of total mutations described for each Sanfilippo syndrome (MPS III) subtype (HGMD Profesional 2020.3; assessed on 9 October 2020).

	Total Mutations	Missense/Nonsense	Small Deletions	Small Insertions	Small Indels	Splicing	Gross Deletions	Gross Insertions and Duplications	Complex Rearrangements
A (SGSH)	155	118	20	9	1	3	3	1	0
B (NAGLU)	229	167	29	16	1	8	4	4	0
C (HGSNAT)	77	43	6	6	1	15	4	1	1
D (GNS)	25	7	5	4	1	4	2	0	2

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
