# Peer review of "Sanfilippo Syndrome: Molecular Basis, Disease Models and Therapeutic Approaches"

_ijms, 2020, doi:10.3390/ijms21217819_

Round 1

Reviewer 1 Report

The paper by Benetò and colleagues is a quite extensive review of the current knowledge of Sanfilippo syndrome pathogenesis and treatment. In general the paper is quite interesting, although many papers have been written on the same topic. I personally think the Authors should have dedicated a bit more on the description of the pathogenesis, by clarifying the contribution of target cellular pathways implicated in the onset of primary tissue defects. There is a quite large part regarding the therapeutic options, but few details of the biological role of HS on target cellular process (just mentioned in Par.2.3).

In my opinion the paper should be corrected in several parts, listed below, before its publication:

  • Line 36, “responsible for heparan sulfate (HS) degradation, an specific GAG”: this sentence is badly written and needs to be corrected.
  • Line 39, 175 and 393. The word “affectation” is incorrect. Please change the term. The Authors have improperly used the term “affect” many times.
  • Line 44-47: this sentence is confusing and need to be rewritten
  • Line 48: aggregation is certainly wrong. Authors should clarify what they would mean.
  • Line 70: “on average”
  • Line 167-168. There is mistake in the sentence. I guess the Authors should divide it into two sentences.
  • Line 184: I would suggest to add” the term “imbalanced turnover” to the term “storage” as FGF and BMP ligands, as the signaling impairment would be certainly affected by disrupted ligands bioavailability or receptor recycling, as previously postulated.
  • Line 266-267: put LMP and GM3 positive before storage vesicles.
  • Line 311: change the term improve. It is incorrect
  • Line 329: change the term “handicaps”
  • Line 290: remove “in the field”
  • Line 391: change the term neurobiological. It is wrong.
  • Line 459 change the term “compound “ into “transducing vector”
  • Line 486: change “Laboratorios del…”. This is not English
  • Line 492-494: use the past tense/past participle

Author Response

The paper by Benetò and colleagues is a quite extensive review of the current knowledge of Sanfilippo syndrome pathogenesis and treatment. In general the paper is quite interesting, although many papers have been written on the same topic. I personally think the Authors should have dedicated a bit more on the description of the pathogenesis, by clarifying the contribution of target cellular pathways implicated in the onset of primary tissue defects. There is a quite large part regarding the therapeutic options, but few details of the biological role of HS on target cellular process (just mentioned in Par.2.3).

We thank the reviewer for considering our manuscript interesting. Although we agree with the reviewer that the parts seem a little bit unbalanced, it is important to note that there have been many more studies in the last years in the field of therapy than in the pathological mechanisms of the disease.

In my opinion the paper should be corrected in several parts, listed below, before its publication:

  • Line 36, “responsible for heparan sulfate (HS) degradation, an specific GAG”: this sentence is badly written and needs to be corrected.

We thank the reviewer for the suggestion. We have now changed this sentence to: “responsible for the degradation of heparan sulfate (HS), a specific GAG” in the new version of the manuscript.

  • Line 39, 175 and 393. The word “affectation” is incorrect. Please change the term. The Authors have improperly used the term “affect” many times.

We thank the reviewer for pointing out this mistake. We have now changed the word affectation in all places in the text by pathology (lines 15, 39, 173, 177, 338, 409).

  • Line 44-47: this sentence is confusing and need to be rewritten

We have now changed this sentence to the following: “For this reason, a better understanding of Sanfilippo syndrome underlying mechanisms can contribute to improve our knowledge on the role of impaired lysosomal function in age-related neurodegenerative disorders.”

  • Line 48: aggregation is certainly wrong. Authors should clarify what they would mean.

We agree with the reviewer that aggregation was the wrong word to use. We meant accumulation and we have now changed it in the new version of the manuscript.

  • Line 70: “on average”

We have now corrected this in the new version of the manuscript.

  • Line 167-168. There is mistake in the sentence. I guess the Authors should divide it into two sentences.

We thank the reviewer for noticing the mistake. We have corrected the sentence as follows: “Importantly, they regulate several signalling pathways, control the proliferative capacity of neural progenitors, are essential for brain patterning and neurogenesis, and crucially participate in the processes of neuronal migration, axon guidance and synaptogenesis.”

  • Line 184: I would suggest to add” the term “imbalanced turnover” to the term “storage” as FGF and BMP ligands, as the signaling impairment would be certainly affected by disrupted ligands bioavailability or receptor recycling, as previously postulated.

We agree with the reviewer´s suggestion and have changed the sentence accordingly in the new version of the manuscript.

  • Line 266-267: put LMP and GM3 positive before storage vesicles.

We have written this sentence as suggested by the reviewer in the new version of the manuscript.

  • Line 311: change the term improve. It is incorrect

This sentence has been removed from the new version of the manuscript.

  • Line 329: change the term “handicaps”

We have now changed this word and used “limitations” in the new version of the manuscript.

  • Line 290: remove “in the field”

We have removed it as suggested by the reviewer.

  • Line 391: change the term neurobiological. It is wrong.

We agree with the reviewer and have changed the term to neurological in the new version of the manuscript

  • Line 459 change the term “compound “ into “transducing vector”

We agree with the reviewer and have changed the text as suggested.

  • Line 486: change “Laboratorios del…”. This is not English

Laboratoris Esteve S.A. is a Catalan company and we used the original name of it. However, considering reviewer’s concern, we have decided to use Esteve Laboratories instead.

  • Line 492-494: use the past tense/past participle

We have now corrected this sentence as follows: “Preliminary data showed a dose-dependent and sustained reduction in cerebrospinal HS after 30 days. In the case of Sanfilippo syndrome types A and B, two clinical trials based on intracerebral injection of AAV have been already completed…”.

Reviewer 2 Report

Beneto and colleagues present a nice review on the Sanfilippo syndrome, where they compile the knowledge on this disorder, with a special focus on the available animal and cellular models that can be used to contribute to increase our understanding on the mechanisms underlying this disorder.

The paper is well organized and easy to read; references are updated and the overall message of the manuscript is clear.

I have no major doubts in recommending it for publication. There are only a few points, which I have listed in the attached document that I would like to see corrected and/or referred to.

Author Response

Beneto and colleagues present a nice review on the Sanfilippo syndrome, where they compile the knowledge on this disorder, with a special focus on the available animal and cellular models that can be used to contribute to increase our understanding on the mechanisms underlying this disorder.

The paper is well organized and easy to read; references are updated and the overall message of the manuscript is clear.

I have no major doubts in recommending it for publication. There are only a few points, which I have listed in the attached document that I would like to see corrected and/or referred to.

We thank the reviewer for the positive general comment on the manuscript.

  • Page 2. Line 54 – the authors state that Sanfilippo syndrome is caused by a “lack” of enzymatic activity. I would suggest that the authors rephrase this sentence in such a way that it implies that the disorder is caused not only by a lack of activity, but also from a deficient activity of any of the relevant enzymes.

We agree with the reviewer’s comment and we have changed now lack by: “Insufficient or complete loss”

  • Page 2, line 59 – ARSG should be written in italics, as it refers to a gene.

We thank the reviewer for pointing this out and have corrected the text accordingly.

  • Page 2, lines 58-60 – I believe it would be interesting to refer whether those mutations in ARSG are present in homozygosity or compound heterozygosity. Are both alleles affected?

We have now specified that patients have “a homozygous mutation in ARSG”. In the study we refer to, patients from one family carrying the same mutations on both alleles were studied. Therefore, both alleles are affected.

  • Page 2, line 70 – I would recommend the authors to include a reference for the Sanfilippo syndrome incidence estimate.

We agree with the reviewer and have included now the new reference number 15 from Khan et al., 2017 Mol Genet Metab (https://pubmed.ncbi.nlm.nih.gov/28595941/).

  • Page 2 and 3, lines 75 to 110 – this is a general comment that applies to the authors summary of all Sanfilippo subtypes. The total numbers of mutations underlying each subtype they refer in the text are never in accordance with those depicted in table 1. That should be corrected.

Still on the subject of disease-causing mutations, I believe table 1 should be updated in such a way that it presents the most up-to-date numbers, which is not the case in the present version of the manuscript. I know disease-causing mutations are constantly being updated. Still, I believe ideally, when the manuscript gets published, it should reflect the most updated numbers.

So, I took the liberty to write them down, thus updating the table.

MPS III subtype

Total mutations

Missense/ nonsense

Small deletions

Small insertions

Small indels

Splicing

Gross deletions

Gross insestions + dup

Complex rearrang

A

155

118

20

9

1

3

3

1

0

B

229

167

29

16

1

8

4

4

0

C

77

43

6

6

1

15

4

1

1

D

25

7

5

4

1

4

2

0

2

This data was gathered from HGMD Professional 2020.3; assessed on 9th October, 2020. Please change the table heading accordingly.

We thank the reviewer for pointing out this mistake and for the update. We have now corrected the table and the numbers in the text to be matched and updated.

  • Page 2, line 88 – NAGLU maps to 2 instead of 17q21.1. That should be corrected.

We thank the reviewer for the correction and have changed it in the new version of the manuscript.

  • Page 4, lines 158 to 162 – I would recommend the authors to include the acronym for each enzyme of the GAG degradation pathway, right after they enumerate them, as follows: “(…) iduronate 2-sulfatase (IDS), α-L-iduronidase (IDUA), heparan N-sulfatase or sulfamidase (SGSH, mutated in Sanfilippo syndrome type A), acetyl-CoA α-glucosaminide N- acetyltransferase (HGSNAT, mutated in Sanfilippo syndrome type C), α-N-acetylglucosaminidase (NAGLU, mutated in Sanfilippo syndrome type B), glucuronate 2-sulfatase (GDS), β-glucuronidase (GUSB) and N-acetylglucosamine 6 sulfatase (GNS, mutated in Sanfilippo syndrome type D)”.

Otherwise, the acronyms appear in the figure without previous reference in the text.

Another possibility would be to include the disease-related acronyms in subsections 2.1 to 2.4.

We agree with the reviewer’s suggestion and have included the acronyms in the new version of the manuscript.

  • Page 9, lines 359 to 361 – why not including a reference to the fact that work was performed by their lab? I understand those results were obtained by a former team member (also co-author of this manuscript), but science is a continuum, and their current studies build upon those first observations. So, I would also refer them as part of the team’s work, which is actually quite remarkable in the field.

We thank the reviewer for the suggestion and have changed the beginning of the sentence as follows: “In another study performed in our lab”.

  • Page 9, lines 388 – the authors should include here the acronym BBB, since it is the first time they refer to the blood brain barrier. Later on, they do use the acronym (line 468) but never disclose its meaning.

We thank the reviewer for pointing this out and have included the acronym in the new version of the manuscript. However, we included it in the line 323, when the blood-brain barrier is mentioned for the first time, and used the acronym posteriorly.

  • Page 10, line 416 – please disclose the meaning of WT. I know it’s quite a standard acronym but still…

We have now included the meaning of WT in the new version of the manuscript.

  • Page 10, lines 418 to 420 – the authors make a brief reference to the fact that HGSNAT is not mannose 6-phosphated. I would recommend them to briefly comment on the role of the M6P tag for lysosomal targeting. They can do it either here or earlier in the manuscript, when presenting the basis of ERT (page 8; section 5.1).

I saw they have briefly discussed this issue on page 11, when referring to gene therapy and their remarks are quite right. I would just recommend this disclosure to be made earlier in the manuscript.

We agree with the reviewer comment and we have performed the following changes in the new version of the manuscript.

At the beginning of ERT: “The success of any therapy relying on administration or production of the correct form of the lysosomal enzyme relies on the fact that these proteins are tagged with mannose 6-phosphate (M6P) for correct trafficking towards the lysosome. Considering that cells have M6P receptors in the membrane, lysosomal enzymes can be endocytosed and arrive to the lysosome to perform their function [71].”

At the end of Stem Cell Therapy: “However, for Sanfilippo C it is important to consider that HGSNAT does not have a M6P tag and is a membrane protein, therefore secretion and uptake of this enzyme by deficient cells may not be successful.”

At the end of gene therapy: “However, as for ERT, gene therapy success for lysosomal enzymes relies in the ability of transduced cells to share the correct lysosomal enzyme through M6P receptors with non-transduced neighboring cells [68]. As mentioned above, HGSNAT is a lysosomal transmembrane protein that does not undergo the M6P pathway. For this reason, Sanfilippo C syndrome might not be the best candidate for gene therapy strategy, although some interesting results have been obtained with a novel AAV [124].”

  • Page 10, line 427 – please disclose the meaning of ALS: amyotrophic lateral sclerosis.

We agree with the reviewer and considering that it is not appearing again in the text, we have written only the full name.

  • Page 10, lines 431 and 432 – this data has already been presented in a previous section of the manuscript (line 376). I would not repeat it. Still, if the authors want to keep this remark, they may as well add a brief introduction such as “as already referred”…

We have now included the brief introduction “as already referred” as suggested by the reviewer.

  • Page 10 and 11 – the authors should choose whether they use the acronym AAVs or AVV, for the plural. Whatever abbreviation form they chose, they should then stick to it thought the manuscript.

Also, in line 494, the authors should replace the full “adeno-associated virus” by the abbreviated form. And, in line 489, “blood-brain barrier” by BBB; in line 456, “heparan sulfate” by HS.

As suggested by the reviewer, we have now used only AAV, and replace full forms by the abbreviated ones in the new version of the manuscript.

General comment:

Even though the overall English language and style are fine, a few minor spell check and rephrasing is required every now and again. Examples:

  • Page 1, line 36-37 – “an specific GAG”

It has been changed to “a specific GAG”

  • Page 5, line 168 – “(…) they regulate several signalling pathways, control the proliferative capacity of neural progenitors, it is essential for brain patterning and neurogenesis…”

It has been changed to “…are essential…”

  • Page 5, line 170 – “The fact that the CNS present…”

It has been changed to “The fact that the CNS has”

  • Page 5, line 176 – “HS accumulation causes an alteration in the lysosomal environment since it can bind to various…”

The sentence has been modified as follows: “HS accumulation causes an alteration in the lysosomal environment since the excess of undegraded molecules can bind to various hydrolases reducing their activity…”

  • Page 10, line 430 – “Gene therapy is the most promising therapeutic options for LSDs”

It has been changed to ““Gene therapy is the most promising therapeutic option for LSDs”.

  • Page 11, line 453 – “AAV5 has been also used in another clinical trial…”

It has been changed to “AAV5 has also been used in another clinical trial…”

Reviewer 3 Report

The authors very well described the storage pathology in Sanfilippo syndrome, describing the impact of GAG accumulation on the development of neurodegenerative processes and the different therapeutic strategies to treat these severe diseases.

Interestingly, they clearly summarized the different steps of biosynthesis and degradation of heparan sulfates in the MPSs and how the storage of these GAGs drives the CNS alterations including the autophagic pathway impairment and inflammation.

Moreover, the authors also underlined the importance to study the CNS pathology mechanism and the possible therapeutic strategies in different animal models and in advanced cellular models like iPSC lines.

They finally thoroughly reviewed all the therapeutic approaches developed for the treatment of CNS pathology in Sanfilippo disorder such as ERT, SRT, small molecules treatments, stem cell therapy and gene therapy underlining the clinical potential of the AAV-mediated gene therapy strategies.

In conclusion, the manuscript is very interesting and offer a precise and clear overview of models and therapeutic protocols for Sanfilippo syndrome.

For all these reasons, it results suitable for publication.

Author Response

The authors very well described the storage pathology in Sanfilippo syndrome, describing the impact of GAG accumulation on the development of neurodegenerative processes and the different therapeutic strategies to treat these severe diseases.

Interestingly, they clearly summarized the different steps of biosynthesis and degradation of heparan sulfates in the MPSs and how the storage of these GAGs drives the CNS alterations including the autophagic pathway impairment and inflammation.

Moreover, the authors also underlined the importance to study the CNS pathology mechanism and the possible therapeutic strategies in different animal models and in advanced cellular models like iPSC lines.

They finally thoroughly reviewed all the therapeutic approaches developed for the treatment of CNS pathology in Sanfilippo disorder such as ERT, SRT, small molecules treatments, stem cell therapy and gene therapy underlining the clinical potential of the AAV-mediated gene therapy strategies.

In conclusion, the manuscript is very interesting and offer a precise and clear overview of models and therapeutic protocols for Sanfilippo syndrome.

For all these reasons, it results suitable for publication.

We thank the reviewer for considering our manuscript very interesting and suitable for publication.